# S2VG: Soft Stochastic Value Gradient method

## Abstract

Model-based reinforcement learning (MBRL) has shown its advantages in sample-efficiency over model-free reinforcement learning (MFRL). Despite the impressive results it achieves, it still faces a trade-off between the ease of data generation and model bias. In this paper, we propose a simple and elegant model-based reinforcement learning algorithm called soft stochastic value gradient method (S2VG). S2VG combines the merits of the maximum-entropy reinforcement learning and MBRL, and exploits both real and imaginary data. In particular, we embed the model in the policy training and learn $Q$ and $V$ functions from the real (or imaginary) data set. Such embedding enables us to compute an analytic policy gradient through the back-propagation rather than the likelihood-ratio estimation, which can reduce the variance of the gradient estimation. We name our algorithm Soft Stochastic Value Gradient method to indicate its connection with the well-known stochastic value gradient method in (Heess et al., 2015).

## 1 Introduction

Reinforcement learning can be generally classified into two categories: model-free reinforcement learning (MFRL) and model-based reinforcement learning (MBRL). The last several years have witnessed the great success of MFRL especially in playing video games, robotic control and motion animation (Mnih et al., 2015; Lillicrap et al., 2015; Schulman et al., 2017; Peng et al., 2018). However, even for some simple tasks, hundreds of millions of samples are required for an agent to learn a good control policy. In many industry scenarios, such as health care and financial services, the algorithm requiring tremendous interactions with the environment is not applicable or too expensive to deploy. To this end, several recent works have advocated the model-based approach, where the higher sample-efficiency is achieved by leveraging the learned dynamics and reward model (Buckman et al., 2018; Feinberg et al., 2018). It generally augments the real data with the data from dynamics models, uses rollout to improve target for temporal difference learning, or directly incorporates the model into the Bellman equation (Luo et al., 2018; Heess et al., 2015). These works have demonstrated promising results on several benchmarks with a small number of interactions with the environment.

Despite its recent success, MBRL still faces a challenging problem, i.e., the model-bias, where the imperfect dynamics model would degrade the performance of the algorithm (Kurutach et al., 2018). Unfortunately, such things always happen when the environment is sufficiently complex. There are a few efforts to mitigate such issue by combining model-based and model-free approaches. Heess et al. (2015) compute the value gradient along real system trajectories instead of planned ones to avoid the compounded error. Kalweit & Boedecker (2017) mix the real data and imaginary data from the model and then train $Q$ function. An ensemble of neural networks can be applied to model the environment dynamics, which effectively reduces the error of the model (Kurutach et al., 2018; Clavera et al., 2018; Chua et al., 2018).

We observe that most recent algorithms with promising results apply Dyna-style update (Sutton, 1990; Kurutach et al., 2018; Luo et al., 2018). They collect real data using current policy to train the dynamics model. Then the policy is improved using state-of-the-art model-free reinforcement learning algorithms with imagined data generated by the learned model. Our *insight* is that why not *directly embed* the model into the policy improvement? To this end, we derive a model-based reinforcement learning algorithm in the framework of the maximum entropy reinforcement learning

(Ziebart et al., 2008). Dynamics model and reward model are trained with the real data set collected from the environment. Then we simply train $Q$ and $V$ function using the real data set with the update rule derived from the maximum entropy principle (several other advanced ways to include the imaginary data can also be applied, see details in section 3). In the policy improvement step, the stochastic actor samples an action with real state as the input, and then the state switches from $s$ to $s'$ according to the learned dynamics model. We link the learned dynamics model, reward model, and policy to compute an analytic policy gradient by the back-propagation. Comparing with likelihood-ratio estimator usually used in MFRL method, such value gradient method would reduce the variance of the policy gradient (Heess et al., 2015). The other *merit* of S2VG is its computational efficiency. Several state-of-the-art MBRL algorithms generate hundreds of thousands imaginary data from the model and a few real samples. Then the *huge* imaginary data set feeds into MFRL algorithms, which may be sample-efficient in terms of real samples but not computational-friendly. On the contrary, our algorithm embeds the model in the policy update. Thus we can implement it efficiently by computing policy gradient several times in each iteration (see our algorithm 1) and do not need to do calculation on the huge imaginary data set.

We name our algorithm soft stochastic value gradient to indicate its connection with SVG (Heess et al., 2015). Notice there are several differences between S2VG and SVG. Firstly, to alleviate the issue of the compounded error, SVG proposes a relatively conservative algorithm where just real data is used to evaluate policy gradients. Thus imaginary data is wasted, even the data from the short rollouts from the model can be trusted to some extent. In our work, the policy is trained with the model and imaginary dataset $m$ times in each iteration of the algorithm. Secondly, we derive our algorithm in the framework of the maximum entropy reinforcement learning. The maximum entropy updates could improve the robustness under the model estimation error (Ziebart et al., 2010). In addition, it encourages the exploration, prevents the early convergence to the sub-optimal policies, and shows state-of-the-art performance in MFRL (Haarnoja et al., 2018). Thirdly, S2VG avoids the importance sampling in the off-policy setting by sampling the action from $\pi$ and transition from $f(s, a)$, which further reduces the variance of the gradient estimation.

#### CONTRIBUTIONS

We derive an elegant, *sample-efficient*, and *computational-friendly* Dyna-style MBRL algorithm in the framework of the maximum entropy reinforcement learning with a principled way. Different from the traditional MBRL algorithm, we directly *embed* the model in the policy improvement, which could reduce the variance in the gradient estimation and avoid the computation on huge imaginary dataset. In addition, since the algorithm is *off-policy*, it is sample-efficient. At the same time, the maximum entropy principle encourages exploration and improves performance, which has been observed in MFRL (Haarnoja et al., 2017). We test our algorithm on several benchmark tasks in Mujoco simulation environment (Todorov et al., 2012) and demonstrate that our algorithm can achieve state-of-the-art performance [1].

## 2 PRELIMINARIES

In this section, we first present some backgrounds on the Markov decision process. Then we introduce the knowledge on the maximum entropy reinforcement learning (Ziebart et al., 2008) and stochastic value gradient (Heess et al., 2015) since parts of them are the building blocks of our algorithm.

### 2.1 MDP

Markov Decision Process (MDP) can be described by a 5-tuple $(\mathcal{S}, \mathcal{A}, \mathcal{R}, \mathcal{P}, \gamma)$: $\mathcal{S}$ is the state space, $\mathcal{A}$ is the action space, $\mathcal{P} = (P(s'|s, a))_{s, s' \in \mathcal{S}, a \in \mathcal{A}}$ are the transition probabilities, $R = (r(s, a))_{s, s' \in \mathcal{S}, a \in \mathcal{A}}$ are the real-valued immediate rewards, and $\gamma \in (0, 1)$ is the discount factor. A policy is used to select actions in the MDP. In general, the policy is stochastic and denoted by $\pi$, where $\pi(a_t|s_t)$ is the conditional probability density at $a_t$ associated with the policy. The state value evaluated on policy $\pi$ could be represented by $V^\pi(s) = \mathbb{E}_\pi[\sum_{t=0}^{\infty} \gamma^t r(s_t, a_t)|s_0 = s]$ on immediate reward return $r = (R(s, a))_{s, s' \in \mathcal{S}, a \in \mathcal{A}}$ with discount factor $\gamma \in (0, 1)$ along the horizon

---

[1] Code is submitted anonymously at https://github.com/S2VG-anonymous1/S2VG

$t$. The state-action value evaluated on policy $\pi$ represent the expected return on the specific action $a$: $Q^{\pi}(s, a) = r(s_0, a_0) + \mathbb{E}_{\pi}[\sum_{t=1}^{\infty} \gamma^t r(s_t, a_t) | s_0 = s, a_0 = a]$.

## 2.2 MAXIMUM ENTROPY REINFORCEMENT LEARNING

Maximum entropy reinforcement learning augments the reward with an entropy term, such that the optimal policy aims to maximize the new reward function at each visited state :

$$\max_{\pi} J(\pi) = \sum_{t=0}^{T} \mathbb{E}_{(s_t, a_t) \sim \rho_{\pi}}[r(s_t, a_t) + \alpha \mathcal{H}(\pi(\cdot|s_t))], \qquad (1)$$

where $\mathcal{H}(\pi(\cdot|s_t))$ is an entropy term scaled by $\alpha$ (Ziebart et al., 2008). The optimal policy in equation 1 can be obtained by the following soft-Q update (Fox et al., 2016),

$$Q(s_t, a_t) \longleftarrow r(s_t, a_t) + \gamma \mathbb{E}_{s_{t+1} \sim p}[V(s_{t+1})] \text{ and } V(s_t) \longleftarrow \alpha \log(\int_{\mathcal{A}} \exp(\frac{1}{\alpha} Q(s_t, a_t)) da_t).$$

Above iterations define the soft $Q$ operator, which is a contraction. The optimal policy $\pi^*(a|s)$ can be recovered by $\pi^{\star}(a_t|s_t) = \frac{\exp(\frac{1}{\alpha} Q^*(s_t, a_t))}{\int_{\mathcal{A}} \exp(\frac{1}{\alpha} Q^*(s_t, a_t)) da_t}$, where $Q^*$ is the fixed point of the soft-Q update. We refer readers to the work (Ziebart et al., 2008; Haarnoja et al., 2017) for more discussions. In soft actor-critic (Haarnoja et al., 2018), the optimal policy $\pi^*(a_t|s_t)$ is approximated by a neural network $\pi_{\theta}(a_t|s_t)$, which is solved by the following optimization problem

$$\max_{\pi_{\theta}(a_t|s_t)} \mathbb{E}_{s_t \sim p(s_t)} \mathbb{E}_{a_t \sim \pi_{\theta}(a_t|s_t)}[Q(s_t, a_t) - \alpha \log \pi_{\theta}(a_t|s_t))].$$

## 2.3 STOCHASTIC VALUE GRADIENT

Stochastic value gradient method is a model-based algorithm which is designed to avoid the compounded model errors by only using the real-world observation and gradient information from the model (Heess et al., 2015). The algorithm directly substitutes the dynamics model and reward model in the Bellman equation and calculates the gradient. To perform the backpropagation in the stochastic Bellman equation, re-parameterization trick is applied to evaluate the gradient on real-world data. The stochastic policy $\pi(a|s; \theta)$ with parameter $\theta$ could be optimized by the policy gradient in the following way

$$\frac{\partial V(s)}{\partial \theta} \approx \mathbb{E}_{\eta, \zeta}[\frac{\partial \hat{r}(s, a)}{\partial a} \frac{\partial \pi(a|s)}{\partial \theta} + \gamma(\frac{\partial V'(s')}{\partial s'} \frac{\partial f(s, a)}{\partial a} \frac{\partial \pi(a|s)}{\partial \theta})], \qquad (2)$$

where $\eta$ and $\zeta$ are the policy and environment re-parameterization noise which could be directly sampled from a prior distribution or inferred from a generative model $g(\eta, \zeta|s, a, s')$. The $f(s, a)$ and $\hat{r}(s, a)$ are dynamics model and reward model respectively.

## 3 SOFT STOCHASTIC VALUE GRADIENT METHOD

In this section, we introduce our algorithm soft stochastic value gradient (S2VG) method. Our objective is to design an off-policy MBRL algorithm under the maximum entropy framework, which could improve the exploration, enhance the training robustness through entropy maximization. Particularly, we optimize the following equation

$$J(\theta) = \sum_{t=0}^{T} \mathbb{E}_{\hat{\rho}_{\pi}(\tau)}[\hat{r}(s_t, a_t) + \mathcal{H}(\pi(a_t|s_t))], \qquad (3)$$

where we omit the regularizer parameter $\alpha$ of the entropy term in the following discussion to ease the exposition. $\hat{\rho}_{\pi} = p(s_0) \prod_{t=0}^{T} f(s_t, a_t) \pi(a_t|s_t)$, $f(s_t, a_t)$ is the learned dynamics model, and $\hat{r}$ is the reward model. We then derive the update rule following the similar step in the probabilistic reinforcement learning (Levine, 2018), which generally includes policy evaluation and policy improvement steps.

Similar to the policy evaluation without the entropy term, we have the soft value function update $V(s_t) = \mathbb{E}_{\pi(a_t|s_t)}[\hat{r}(s_t, a_t) - \log \pi(a_t|s_t) + \gamma \mathbb{E}_{s \sim f} V(s_{t+1})]$. For convenience, we define the $Q$ function in the following,

$$Q(s_t, a_t) = \hat{r}(s_t, a_t) + \gamma \mathbb{E}_{s_{t+1} \sim f}[V(s_{t+1})]. \tag{4}$$

The value function update can be reformulated as

$$V(s_t) = \mathbb{E}_{\pi(a_t|s_t)}[Q(s_t, a_t) - \log \pi(a_t|s_t)]. \tag{5}$$

The optimal policy (policy improvement step) at each step $t$ is $\pi(a_t, s_t) = \frac{\exp(Q(s_t,a_t))}{\int_a \exp(Q(s_t,a_t)da_t}$. The derivation is almost the same with (Levine, 2018; Haarnoja et al., 2018), expect that we use learned dynamics model and reward function here. Notice this optimal policy can be approximated by a parametric function $\pi_\theta(a_t|s_t)$ and obtained by solving $\max_{\pi_\theta(a_t|s_t)} \mathbb{E}_{s_t \sim p(s_t)} \mathbb{E}_{a_t \sim \pi_\theta(a_t|s_t)}[Q(s_t, a_t) - \log \pi_\theta(a_t|s_t))]$. However such way used in the MFRL can not leverage the model information. We leave the our derivation and discussion on the policy improvement in section 3.3.

## 3.1 MODEL LEARNING

The transition dynamics and rewards could be modeled by non-linear function approximations as two independent regression tasks which have the same input but different output. Particularly, we train two independent deep neural networks with parameter $\omega$ and $\varphi$ to represent the dynamics model $f$ and reward model $\hat{r}$ respectively. To better represent the stochastic nature of the dynamic transitions and rewards, we implement re-parameterization trick on both $f$ and $\hat{r}$ with input noises $\zeta_\omega$ and $\zeta_\varphi$ sampled from Gaussian distribution $\mathcal{N}(0,1)$. Hence, networks would generate mean: $\mu_\omega$, $\mu_\varphi$, and variance: $\sigma_\omega$, $\sigma_\varphi$ separately and compute the result by $\mu_\omega + \sigma_\omega \zeta_\omega$ and $\mu_\varphi + \sigma_\varphi \zeta_\varphi$, respectively.

Above two models could be optimized by sampling the data from the (real data) replay buffer $\mathcal{D}$ and minimizing the mean square error:

$$J(\omega) = \frac{1}{2}\mathbb{E}_{\mathcal{D},\zeta_\omega}[(f(s,a) - s')^2], J(\varphi) = \frac{1}{2}\mathbb{E}_{\mathcal{D},\zeta_\varphi}[(\hat{r}(s,a) - r)^2]. \tag{6}$$

This supervised learning problem could be solved by off-the-shelf optimizers such as Adam optimizer (Kingma & Ba, 2014). Other techniques could also be leveraged to reduce the risk of overfitting in the limited data set such as adding dropout layers, performing early stopping, and training through ensemble models.

## 3.2 VALUE FUNCTION LEARNING

Equation 4 and equation 5 define model-based and model-free policy evaluation steps respectively. Equation 4 would introduce the model error from the model dynamics into the value estimation and produce biased results. To avoid this model error, we update $Q$ with the real transition $(s, a, r, s')$ from the real data replay buffer. Such update is non-biased but may suffer from high variance under the low-data regime. Therefore, we leverage the value expansion (Feinberg et al., 2018) to balance the bias and variance by using both real-world data and imaginary rollout. If $Q$ function and $V$ function are parameterized by $\phi$ and $\psi$ respectively, they could be updated by minimizing the new objective function with the value expansion on imaginary rollout:

$$J(\phi) = \frac{1}{H} \sum_{t=0}^{H-1} \left(Q_\phi(\hat{s}_t, \hat{a}_t) - \left(\sum_{k=t}^{H-1} \gamma^{k-t}\hat{r}_k + \gamma^{H-t}V_\psi(\hat{s}_H))\right)\right)^2, \tag{7}$$

where only the initial tuple $\tau_0 = (\hat{s}_0, \hat{a}_0, \hat{r}_0, \hat{s}_1)$ is sampled from replay buffer $\mathcal{D}$ with real-world data, and later transitions are sampled from the imaginary rollout from the model. H here is the time step of value expansion using real data. $\tau$ is the training tuple and $\tau_0$ is the initial training tuple. Note that when $H = 1$, it reduces to the case where just real data is used.

The $V$ function is learned by minimizing the following error

$$J(\psi) = \mathbb{E}_{s_t \sim \mathcal{D}}[\frac{1}{2}(V_\psi(s_t) - \mathbb{E}_{a_t \sim \pi}[Q_\phi(s_t, a_t) - \log \pi(a_t|s_t)])^2]. \tag{8}$$

Notice the training of $V$ function is only on the real data set $D$. In our experimental section, we do ablation study on $H$. Interestingly, we find that the case with $H = 1$ has the best result. One possible explanation is that we have already embedded model in the policy update. Hence, including the imaginary data in value function learning would mislead directions of policy gradients.

### 3.3 POLICY LEARNING

Then we consider the policy improvement step, i.e., to calculate the optimal policy at each time step. One *naive* way is to optimize the following problem $\max_{\pi(a_t|s_t)} \mathbb{E}_{s_t\sim p(s_t)}\mathbb{E}_{a_t\sim\pi(a_t|s_t)}[Q(s_t, a_t) - \log \pi(a_t|s_t))]$ as that in MFRL (Levine, 2018; Haarnoja et al., 2018). However, such way cannot leverage the learned dynamics model and reward model. To incorporate the model information, notice that $V(s_t) = \mathbb{E}_{a\sim\pi(a_t|s_t)}[Q(s_t, a_t) - \log \pi(a_t|s_t)]$, thus the policy improvement step is equal to

$$\max_{\pi(a_t|s_t)} \mathbb{E}_{s_t\sim p(s_t)} V(s_t). \tag{9}$$

In the following, we connect the dynamics model, reward model, and value function together by the soft Bellman equation. To begin with, we re-parameterize the dynamics model and policy. Particularly, we set $a = \kappa(s, \eta; \theta)$ and the dynamics model $s' = f(s, a, \zeta)$ for noise variables $\eta \sim \rho(\eta)$ and $\zeta \sim \rho(\zeta)$, respectively. Now we can write the soft Bellman equation in the following way.

$$V(s) = \mathbb{E}_\eta[\hat{r}(s, \kappa(s, \eta; \theta)) - \log \pi(a|s) + \gamma\mathbb{E}_\zeta V'(f(s, \kappa(s, \eta; \theta), \zeta))] \tag{10}$$

To optimize equation 9 and leverage gradient information of the model, we sample $s$ from the real data replay buffer $D$ and take the gradient of $V(s)$ w.r.t. $\theta$

$$\mathbb{E}_{s\sim D}\frac{\partial V(s)}{\partial\theta} = \mathbb{E}_{s\sim D,\eta,\zeta}[\frac{\partial\hat{r}}{\partial a}\frac{\partial\kappa}{\partial\theta} - \frac{1}{\pi}\frac{\partial\pi}{\partial\theta} + \gamma(\frac{\partial V'(s')}{\partial s'}\frac{\partial f}{\partial a}\frac{\partial\kappa}{\partial\theta})]. \tag{11}$$

The equation 11 demonstrates an interesting connection between our algorithm and SVG. Compared with the policy gradient step taken by SVG(1) algorithm (Heess et al., 2015), equation 11 includes one extra term $-(1/\pi)(\partial\pi/\partial\theta)$ to maximize the entropy of policy. We drop importance sampling weights by sampling from the current policy. Also notice that the transition from $(s, a)$ to $s'$ is sampled from the learned dynamics model $f$, while the SVG(1) just utilizes the real data. Thus in the algorithm we can update policy several times in each iteration to fully utilized the model rather than just use the real transition once.

### 3.4 S2VG ALGORITHM

We summarize our S2VG in Algorithm 1. At the beginning of each step, we train dynamics model $f$ and reward model $\hat{r}$ by minimizing the $L_2$ loss shown in equation 6. Then the agent interacts with the environment and stores the data in the real data replay buffer $D$. Actor samples $s_k$ from $D$ and collect $s_{k+1}$ according to the dynamics model $f(s_k, a_k)$. Such imaginary transition is stored in $D_{img}$. Then we train $Q$, $V$ and $\pi$ according to the update rule in section 3. Similar to other value-based RL algorithms, our algorithm also utilizes two $Q$ functions to further reduce the overestimation error by training them simultaneously with the same data but only selecting the minimum target in value updates (Fujimoto et al., 2018). We use the target function for $V$ like that in deep Q-learning algorithm (Mnih et al., 2015), and update it with an exponential moving average. We train policy using the gradient in equation 11. Remark that our $s'$ is sampled from the dynamic model $f(s, a)$, while in SVG, it uses the true transition. Indeed there is a bias and variance trade-off. True transition is unbiased but may have high variance due to the small scale of the data set. Here we claim that one-step rollout of the model is still accurate. Some remarks on the algorithm are in the order. In our implementation, we choose $H = 1$ in equation 7 generally, since we find S2VG with $H = 1$ has the best result in our ablation study (see section 5.3) . In each iteration, we update policy several times, so that the algorithm can utilize the data generated from the model.

## 4 RELATED WORK

There are a plethora of works on MBRL. They can be classified into several categories depending on the way to utilize the model, to search the optimal policy or the function approximator of the

---

**Algorithm 1** Soft Stochastic Value Gradient method

---

**Inputs**: Replay buffer $\mathcal{D}$, imaginary replay buffer $\mathcal{D}_{img}$, policy $\pi_\theta$, value function $V_\psi$, target value function $V_{\bar{\psi}}$. Two $Q$ functions with parameters $\phi_0$ and $\phi_1$, dynamic model $f$ with parameter $\omega$, and reward model $\hat{r}$ with parameter $\varphi$

**for** each iteration **do**

    **1. Train the dynamics model and reward model**

    Calculate the gradients $\nabla_\omega J(\omega), \nabla_\varphi J(\varphi)$ using equation 6 with $\mathcal{D}$, update $\omega$ and $\varphi$

    **2. Interact with environment**

    Sample $a_t \sim \pi(a_t|s_t)$, get reward $r_t$, and observe the next state $s_{t+1}$

    Append the tuple $(s_t, a_t, r_t, s_{t+1})$ into $\mathcal{D}$

    **3. Update the actor, critics $m$ (typically 3 to 5) times**

    Empty $\mathcal{D}_{img}$

    Sample $(s_0, a_0, r_0, s_1') \sim \mathcal{D}$

    **for** each imaginary rollout step $k$ **do**

        Sample $a_k \sim \pi(a_k|s_k)$, get reward $r_k = \hat{r}(s_k, a_k)$, and sample $s_{k+1} \sim f(s_k, a_k)$

        Append the tuple $(s_k, a_k, r_k, s_{k+1})$ into $\mathcal{D}_{img}$

    **end for**

    Calculate the gradient $\nabla_\phi J(\phi)$ using equation 7 with $\bar{\psi}$ and $\mathcal{D}_{img}$

    Calculate the gradient $\nabla_\psi J(\psi)$ using equation 8 with $\mathcal{D}$

    Calculate the gradient $\nabla_\theta V(s)$ using equation 11 with $\mathcal{D}$.

    Update $\phi$, $\psi$, and $\theta$, update $\bar{\psi}$ with Polyak averaging

**end for**

---

dynamics model. Iterative Linear Quadratic-Gaussian (iLQG) (Tassa et al., 2012) assumes that the true dynamics are known to the agent. It approximates the dynamics with linear functions and the reward function with quadratic functions. Hence the problem can be transferred into the classic LQR problem. In Guided Policy Search (Levine & Koltun, 2013; Levine & Abbeel, 2014; Finn et al., 2016), the system dynamics are modeled with the time-varying Gaussian-linear model. It approximated the policy with a neural network $\pi$ by minimizing the KL divergence between iLQG and $\pi$. A regularization term is augmented into the reward function to avoid the over-confidence on the policy optimization. Nonlinear function approximator can be leveraged to model more complicated dynamics. Deisenroth & Rasmussen (2011) use Gaussian processes to model the dynamics of the environment. The policy gradient can be computed analytically along the training trajectory. However, it may suffer from the curse of dimensionality which hinders its applicability in the real problem. Recently, more and more works incorporate the deep neural network into MBRL. Heess et al. (2015) model the dynamics and reward with neural networks, and compute the gradient with the true data. Richards (2005); Nagabandi et al. (2018) optimize the action sequence to maximize the expected planning reward along with the learned dynamics model and then the policy is fine-tuned with TRPO. Luo et al. (2018); Chua et al. (2018); Kurutach et al. (2018) use the current policy to gather the data from the interaction with the environment and then learn the dynamics model. In the next step, the policy is improved (trained by the model-free reinforcement learning algorithm) with a large amount of imaginary data generated by the learned model. Ensemble learning can also be applied to further reduce the model error.

## 5 EXPERIMENTAL RESULTS

In this section, we would like to answer two questions: (1) How does S2VG perform on some benchmark reinforcement learning tasks comparing with other state-of-the-art model-based and model-free reinforcement learning algorithms? (2) How many imaginary data we should use in the value function update?

### 5.1 ENVIRONMENT

To answer these two questions, we do experiment in Mujoco simulation environment (Todorov et al., 2012): InvertedPendulum-v2, HalfCheetah-v2, Reacher-v2, Hopper-v2, Swimmer-v2,Walker2d-v2. Each experiment is tested on five trailed using five different random seeds and initialized parameters. The details of the tasks and experiment implementations can be found in appendix A.

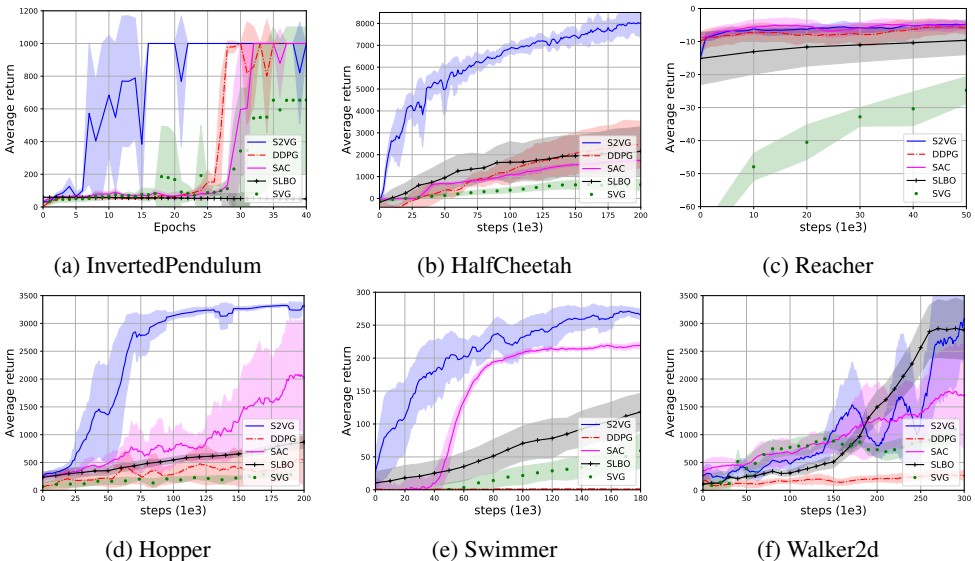

(a) InvertedPendulum     (b) HalfCheetah     (c) Reacher

(d) Hopper     (e) Swimmer     (f) Walker2d

Figure 1: Performance of S2VG with $H = 1$ and other baselines in benchmark tasks. The x-axis is the training step (epoch or step). Each experiment is tested on five trailed using five different random seeds and initialized parameters. For a simple task, i.e., InvertedPendulum, we limit the training steps at 40 epochs. For the other three complex tasks, the total training steps are 200K or 300K. The solid line is the mean of the average return. The shaded region represents the standard deviation. On Invertedpendulum,HalfCheetah, Hopper, Swimmer, S2VG outperforms the other baselines significantly. In the task Walker2d, SLBO is slightly better than S2VG. They both surpass other algorithms. On Reacher, S2VG and SAC perform best.

## 5.2 COMPARISON TO STATE-OF-THE-ART

We compare our algorithm with state-of-the-art model-free and model-based reinforcement learning algorithms in terms of sample complexity and performance. DDPG (Lillicrap et al., 2015) and Soft actor-critic (Haarnoja et al., 2018) are two model-free reinforcement learning algorithms on continuous action tasks. Soft actor-critic has shown its reliable performance and robustness on several benchmark tasks. Our algorithm also builds on the maximum entropy reinforcement learning framework and benefits from incorporating the model in the policy update. Two model-based reinforcement learning baselines are SVG (Heess et al., 2015) and SLBO (Luo et al., 2018). Comparing with SVG, our work avoids the importance sampling and utilizes the maximum entropy principle. Notice in SVG, the algorithm just computes the gradient in the real trajectory, while our S2VG updates policy using the imaginary data $m$ times generated from the model. SLBO is a model-base algorithm with performance guarantees that applies TRPO (Schulman et al., 2015) on the data set generated from the rollout of the model.

Notice that in our implementation, we do *not* use any ensemble learning (Chua et al., 2018) or uncertainty estimation (Malik et al., 2019) on the model. These techniques are known to reduce the model biased. We do not use distributed RL either to accelerate the training. We believe that above-mentioned skills are orthogonal to our work and could be integrated into the future work to further improve the performance. Also for the fairness, we just compare this pure version of S2VG with other baselines. We also notice that some recent works in MBRL modify the benchmarks to shorten the task horizons and simplify the model problem (Kurutach et al., 2018). On the contrary, we test our algorithm in the full-length tasks. In all experiment, we implement S2VG with $H = 1$ and will do ablation study on $H$ in the next section.

We present experimental results in Figure 1. In a simple task, invertedPendulum, S2VG achieves the asymptotic result just using 16 epochs. In HalfCheetah, S2VG's performance is at around 8000 at 200k steps, while all the other baselines' performance is below 2500. In Reacher, S2VG and SAC has similar performance. Both of them are better than other algorithms. In Hopper, the final performance of S2VG is around 3300. The runner-up is SAC whose final performance is around

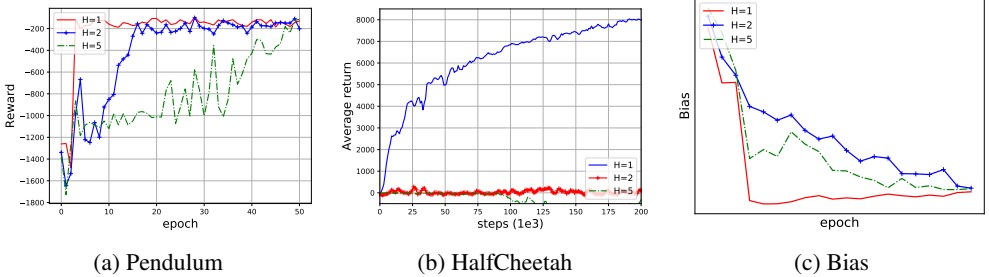

(a) Pendulum

(b) HalfCheetah

(c) Bias

Figure 2: S2VG with value expansion. We do the ablation study on (a) Pendulum and (b) HalfCheetah problem where the x-axis is the training step and the y-axis is the reward. (c) reflects the bias of the value function in the training procedure.

2000. In swimmer, the performance of S2VG is best.In walker2d, SLBO is slighter better than S2VG. Both of them achieve the average return 2900 at 300k timesteps.

## 5.3 DESIGN EVALUATION

In this section, we make the ablation study to understand how much imaginary data we should include in the algorithm. Remark that in our algorithm, the model is embedded in Soft Bellman equation in the policy update step, which means we fully trust the model to compute the policy gradient. While in the value function update, we can either train $Q$ and $V$ using the true data or the data from imaginary rollout in equation 7. In section 5.2, we apply a relatively conservative way, i.e., train $Q$ and $V$ with true data set, i.e., $H = 1$ in equation 7. In the following experiment, we test the algorithm with value expansion, particularly with horizon $H = 2$ and $H = 5$. Our conclusion is that including the imaginary data to train the value function in our algorithm would hurt the performance, especially in complex tasks.

We demonstrate the performance of S2VG with value expansion in Figure 2. We first test the algorithm on a simple task pendulum from OpenAI gym (Brockman et al., 2016) and show the result in Figure 2a. S2VG with $H = 1$ converges to the optimal policy within several epochs. When we increase the value of $H$, the performance decreases. The agent with $H = 5$ just starts to learn the optimal policy at 50 epochs. Then we evaluate the performance of value expansion in a complex task HalfCheetach from Mujoco environment (Todorov et al., 2012). In this task, value expansion with $H = 2$ and $H = 5$ does not work at all. The reason would be that the dynamics model of HalfCheetah introduces more significant model bias comparing to the simple task pendulum. Thus training both policy and value function in the imaginary data set may cause a large error in policy gradient. In Figure 2c, we plot the bias of the value function in the training procedure of Figure 2a. We evaluate the value estimation by averaging the estimated value along 100 states sampled from the replay buffer. Then, we perform Monte Carlo sampling starting from each sampled states with 50 trials and average the discounted return as the true value estimation. We compared the value estimation results of our proposed methods with or without utilizing value expansion. Comparing with the S2VG with $H = 1$, S2VG with $H = 2$ and $H = 5$ introduce more value estimation bias in the learning procedure.

## 6 CONCLUSION AND FUTURE WORKS

In this paper, we propose a new model-based algorithm to directly incorporate models in the policy improvement step. Comparing with the existing method, our algorithm is both sample-efficient and computational-friendly. We test our S2VG on several benchmark tasks and achieve state-of-the-art performance. We can integrate the existing techniques such as ensemble learning and distributed reinforcement learning to further improve the learning speed of S2VG in the future work, since they are orthogonal to our core idea.

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

## A  ENVIRONMENT OVERVIEW AND HYPERPARAMETER SETTING

In this section, we provide an overview of simulation environment in Table 1. The hyperparameter setting for each environment is shown in Table 2.

| Environment Name | Observation Space Dimension | Action Space Dimension | Horizon |
|---|---|---|---|
| Pendulum | 3 | 1 | 200 |
| InvertedPendulum | 4 | 1 | 1000 |
| HalfCheetah | 17 | 6 | 1000 |
| Hopper | 11 | 3 | 1000 |
| Walker2D | 17 | 6 | 1000 |

Table 1: The observation space dimension, action space dimension, and horizon for each simulation environment implemented in the experiment and ablation study.

| | Pendulum | InvertedPendulum | HalfCheetah | Hopper | Walker2D |
|---|---|---|---|---|---|
| Epoch | 50 | 40 | 200 | | 300 |
| Policy Learning Rate | 0.0003 | | | | |
| Value Learning Rate | 0.0003 | | 0.001 | 0.001 | 0.0003 |
| Model Learning Rate | 0.0003 | | | | 0.0001 |
| Alpha value (in entropy term) | 0.2 | 0.1 | 0.4 | | |
| environment steps per epoch | 1000 | | | | |
| Value and Policy Network Architecture | (256,256) | | | | |
| Model Network Architecture | (32,16) | | (256,128) | | (256,256) |
| Train Actor-critic Times ($m$) | 5 | 1 | 5 | | 3 |

Table 2: The hyperparameter used in training S2VG algorithm for each simulation environment. The number in policy, value, and model network architecture indicate the size of hidden units in each layer of MLP. The ReLu activation function is implemented in all architecture.

