# OpenReview forum: "S2VG: Soft Stochastic Value Gradient method"
_ICLR.cc/2020/Conference — Reject_

### Official Review · AnonReviewer3 · 2019-10-22
**Official Blind Review #3**

**Rating:** 1

**Review:**

#rebuttal response
Thanks for the explanations for the difference between S2VG and SVG, new results on two more MuJoCo environments. However, I still think that the motivation of the paper is not clear: the authors may have a good algorithm. But the value of adding entropy on SVG or training the policy on imagination data is not well justified.


#review
This paper proposes a new model-based method that combines the maximum-entropy objective and stochastic value gradient methods. Based on this method, the authors present the Soft Stochastic Value Gradient method (S2VG). Experimental results show that S2VG beats SVG, SAC, DDPG, and SLBO on three MuJoCo tasks.

First of all, the paper lacks a good motivation. SVG already directly embed the model into policy improvement. Then the novelty of this paper is adding an entropy regularizer to improve the robustness under the model estimation error. However, this point is not clarified in the paper. As shown in  Figure 1(a),(d), the performance of S2VG is not stable. Thus I do not find the value of adding a policy entropy regularizer on the SVG method.

Secondly, the comparison is not fair. S2VG should be compared with STEVE ((Buckman et al., 2018), the state of the art model-based method. That is, the claim that S2VG beats state-of-the-art model-based methods is not appropriate.

Finally, the performance improvement is somewhat weak as S2VG only beats other baselines on three MuJoCo tasks.

Question:

(1) Why is the recursive value gradient missing in Eq. (2)?
(2) It is interesting that S2VG outperforms SVG significantly. Does the improvement come from better exploration?
(3) In Figure 2(c), the bias of H=2 is larger than that of H=5. is there any explanation?

**Experience Assessment:**

I have published one or two papers in this area.

**Review Assessment: Checking Correctness Of Derivations And Theory:**

I carefully checked the derivations and theory.

**Review Assessment: Checking Correctness Of Experiments:**

I carefully checked the experiments.

**Review Assessment: Thoroughness In Paper Reading:**

I read the paper thoroughly.

---

> ### Author Response · Authors · 2019-11-15
> **Responds to the comments**
>
> Comments: "First of all, the paper lacks a good motivation. SVG already directly embed the model into policy improvement. Then the novelty of this paper is adding…"
>
> Responds:The reviewer may misunderstand the main contribution of this paper. Although SVG embeds the model in the policy improvement, our way to use the model is quite different from SVG. Let me briefly explain the difference which has been discussed in [section 1 paragraph 4].
>
> Although SVG embeds the model, it does not use the data generated from the model, which is very conservative. This is a quite important difference from S2VG. In general, model-based RL is more sample-efficient since it can utilize the imaginary data. Our method utilizes imaginary data and the update rule is different from SVG. Furthermore, notice that S2VG avoids the importance sampling by sampling the action from the policy and getting the transition data w.r.t. the learned model. In SVG, it has to introduce the importance sampling which may have large variance.
> Our claim is supported by the experimental result shown in the Figure 1. We achieved a huge improvement compared with SVG algorithm. We also compare our algorithm with state-of-the-art off-policy method and demonstrate the superiority of our algorithm .
>
> Comments:" Secondly, the comparison is not fair..."
> Response:There are many state-of-the-art algorithms in reinforcement learning domain. We test the proposed S2VG algorithm with other four baselines which are representative in each domain. DDPG is a common baseline of model-free off-policy RL algorithm with deterministic action. SAC is a state-of-the-art model-free off-policy RL algorithm with stochastic action. SLBO is a state-of-the-art model-based off-policy RL algorithm published in 2018 (Luo, et al., 2018) with performance guarantee. We noticed that the STEVE (Buckman et al., 2018) approach utilized ensemble methods to improve the performance of the model estimation. Hence, it is hard to compare with other RL baseline methods. To ensure the fairness of comparison, all ensemble methods are not used in the experiment, only the “pure” version of S2VG algorithm was tested. In the future work, we would include the ensemble learning in S2VG
>
> Comments: "Finally, the performance improvement is somewhat weak as S2VG only beats other baselines on three MuJoCo tasks."
> Response:  we add experiments on swimmer and reacher. We see our algorithm performs best among five of them. We agree that SLBO is slightly better than our algorithm in walker2d. However, it may require too much on an algorithm to beat all algorithm in all simulation environment.
>
> Comments: "Why is the recursive value gradient missing in Eq. (2)? "
> Response:In Equation 2, we describe the value gradient in infinite horizon. The recursive value gradient term is only shown in the finite horizon. In SVG(1) algorithm (see Heess et al., 2015) which is used in infinite horizon, it didn’t consist any recursive value gradient term in value gradient calculation.
>
>
> Comments:
> "It is interesting that S2VG outperforms SVG significantly. Does the improvement come from better exploration? "
>
> Response: Yes, the S2VG outperforms SVG significantly. The improvement comes from three perspectives. First we trust the short-horizon imaginary rollout and substitute them into the policy improvement. However, the SVG algorithm only use real data in policy improvement. Second, our algorithm an off-policy algorithm without performing importance sampling. Third, our algorithm is constructed based on the maximum entropy RL framework which is used to encourage the exploration and encounter the model error. We clearly describe this statement in Section 1, Paragraph 4:” Firstly, to alleviate the issue of the compounded error, SVG proposes a relatively conservative….”
>
> Comments:
> (2)	In Figure 2(c), the bias of H=2 is larger than that of H=5. is there any explanation?
>
> Response:
> The value expansion is a value estimation technique which seeks the trade-off between bias and variance using both real and imaginary rollout. If H=1, only the real data could be used in the value estimation update. As we know, if only the real data is used in low-data regimes, the evaluation results are unbiased but with high variance. If we use the imaginary rollout, the results are biased but with low variance. Therefore, if we increase the rollout horizon, the bias should increase which is contrary to the experimental results shown in the Figure 2. However, during the test, value expansion rollout is not the only factor that effects the bias of estimation. The non-linear function approximation stochasticity and different training samples per batch caused by different rollout horizons may also contribute to the biased results. In the experiment, we demonstrate the true result from the experiment. The bias of H=2 is larger than the H=5 is not contrary to our decision in using H=1 because the bias of H=1 is the smallest one.

---

### Official Review · AnonReviewer2 · 2019-10-23
**Official Blind Review #2**

**Rating:** 1

**Review:**

This paper presents a slightly new model-based reinforcement learning algorithm, claiming that the new method is elegant and combines the merits of earlier methods. In particular, it claims that the new method is more sample efficient and computationally efficient than previous methods.The novelty of the new method has to do with how the model participates in “the policy improvement step”. The paper does not achieve a high level of quality in its presentation or empirical results, as i discuss further below, and as a result I do not consider it to make a scientific contribution, and I recommend rejection.

The presentation of the method is not clear; it uses an informal notation and leaves out many steps; in many cases the statements are not formally correct. I believe that many of these weaknesses are not introduced in this work but are present also in the works that it builds on. It is possible that someone intimately familiar with the previous works would be able to fully understand this method, but perhaps not, and certainly it does not stand alone.

The new method is presented in Section 3. Equation (3) defines an objective for the policy parameter. It uses things called r-hat, \rho-hat, p, and f. It is not clear whether these things are functions, random variables, or distributions. What is the domain of f? Is r(s,a) a random variable? Is r(s1,a1) different from r(s2,a2) if s1=s2 and a1=a2? Do they have the same distribution? This is just scratching the surface. Many, many things are not well defined, and some critical things, like V, appear to be defined multiple times.

The new method is ultimately presented in Algorithm 1, but it is just a sketch, with essential things left out. What is an “iteration”? Where are the “epochs” discussed in the empirical results? Many steps ask us to calculate gradients with respect to objective functions given by equations in the text. Almost all of these are unclear. To literally calculate the gradient of the expected values would involve a full sweep over the replay buffer, which appears never to be emptied, and thus would grow without bound. It this really what is meant.

There are also many problems with the empirical results. To begin with, there is no discussion of how the many hyper-parameters were set. And the statistics are not present.  The results present means and standard deviations. The paper does not appear to say how many runs were done, but from the wide variation in the standard deviations, it would appear that there were not many. The very basics of a valid experiment are thus missing. We really should not conclude anything from such results, but if they are published, then many readers would conclude things. Our field should not go any further this way, in my opinion.


**Experience Assessment:**

I have published in this field for several years.

**Review Assessment: Checking Correctness Of Derivations And Theory:**

I assessed the sensibility of the derivations and theory.

**Review Assessment: Checking Correctness Of Experiments:**

I assessed the sensibility of the experiments.

**Review Assessment: Thoroughness In Paper Reading:**

I read the paper thoroughly.

---

> ### Author Response · Authors · 2019-11-15
> **Responds to the comments**
>
> With all due respect, I do not agree with most of comments from the reviewer, especially the ones on our notation of reward, dynamics and description of algorithm and experiment.
>
> Let me first explain the contribution of this paper and then answer the detailed questions in reviewer’s comments.
> As the reviewer mentioned, the “policy improvement step” is one of our contribution. Thus let me explain why our policy improvement step is much better than other algorithms briefly since it has been discussed in [section 1, paragraph 3,4]. Comparing with SVG, S2VG uses imaginary data generated from the model. The amount of imaginary data is much larger than the real one, which is also why most model-based RL can reduce the numbr of interaction with the environment. At the same time, S2VG avoids the importance sampling in SVG. As we know, the importance sampling may introduce large variance. Comparing with other model-based RL, our algorithm is more sample-efficient. The reason is quite straightforward and discussed in [section 1, paragraph 3]. Most state-of-the-art model-based RL uses the imaginary data in an “indirect” way, while we embed the model in the policy update. Our experiment also affirms above claims.
>
> Comments: "The new method is presented in Section 3. Equation (3) defines an objective for the policy parameter. It uses things called r-hat, \rho-hat, p, and f. It is not clear..."
> Response： We clearly defined the \r-hat and f as reward model and dynamic model respectively. In Section 2.3:” The f(s,a) and \r-hat(s,a) are dynamics model and reward model respectively.” In Section 3:” f(s_t,a_t) is the learned dynamics model, and r is the reward model.” We also clearly defined the usage of reward model and dynamic model at Section 3.1:” The transition dynamics and rewards could be modeled by non-linear function approximations as ...” Therefore, they are generative models and could generate results by inputting state and action.  We believe that our statement is accurate, especially for domain experts.
>
> In Section 2.1, we clearly define the t in s_t and a_t as time horizon:” with discount factor $gamma\in(0,1)$ along the horizon t”. So, s1 and s2 are different state observation at different time step t=1 and t=2. Notice we do not define V multiple times. V is the expectation of long-term reward (or reward+ entropy term in maximum entropy RL) from the given state. It can also be rewrite in the Bellman equation form as that in section 3, which is quite straightforward for RL community.
> We believe that the statements and definitions on r, f, s, and a are the basic knowledge for all researchers in reinforcement learning research field and therefore don’t need to be introduced with lengthy definition. In fact, we indeed introduced the basic definition of MDP and other abbreviations in Section 2.1.
>
> Comments: "The new method is ultimately presented in Algorithm 1, but it is just a sketch..."
> Response： The iteration means the repetition of a process. In Algorithm 1:”For each iteration do:” means we need to perform the later process iteratively, including three pseudo steps shown in the Algorithm 1. In Figure 1, we clearly show that the ‘epoch’ has the same scale of 1e3 training steps. In the Appendix A, we indicated the environment steps of each epoch!
>
> The gradient would be calculated once and used to update the function approximation which is clearly shown in the Algorithm 1:” Update $\phi$,...” Based on the definition of each objective function in Section 3.2, Section 3.3, we clearly indicated that the data is sampled from replay buffer and is not required to empty the replay buffer after each gradient calculation” Above two models could be optimized by sampling the data from the (real data) replay buffer D and minimizing the mean square error:”, “where only the initial tuple $\tau_0$= is sampled from replay buffer D with real-world data, and later transitions are sampled from the imaginary rollout from the model.”
> In the reinforcement learning domain, we believe that the replay buffer refers to the dataset that the data should be sampled from. This definition is used in many other reinforcement learning publications and should be considered as a common basic knowledge in machine learning domain.
>
> Comments:“There are also many problems with the empirical results. To begin with, there is no discussion of how the many hyper-parameters were set… “
> Response: We have already indicated the parameter setting in the Appendix A in our original version of this paper. Also, we upload our code and encourage all the readers and researchers to investigate our new algorithm. We agree that the trials should be introduced in the paper. We tested and plotted Figure 1 and Figure 2 with 5 trials of running.

---

### Official Review · AnonReviewer1 · 2019-10-23
**Official Blind Review #1**

**Rating:** 3

**Review:**

This paper proposes an off-policy model-based reinforcement learning approach. The proposed paper combines several techniques together. The general framework fits a maximum entropy reinforcement learning, and the policy and q-function updates are formulated in a soft actor-critic manner. To leverage the learned dynamics, the proposed approach adopts value expansion to substitute the learned reward function and forward dynamics function in the Bellman equation for the value function. Also, the imagined roll-out could be used to update the q-function.

I feel the novelty of this approach is a bit limited, considering that the novelty mainly comes from embedding a more powerful maximum entropy RL formulation upon the value expansion [Freinberg et al., 2018] and stochastic value gradient[Heess et al., 2015], rather than from the perspective of model-based modeling. Even though the experiment result is still very promising and it's good to see the method could outperform SLBO in 3 out of 4 cases.


Some other comments:
- The presentation in Sec 3.2 is not clear. Hard to follow without looking into the MVE paper. The definition of \tau and H should be given at that place. Also, the authors claim Q is updated with real *transition*, but later part states only initial tuple is from real data.
- Up to how many roll-out step k could the model reliably work upon should be clearly stated.
- For Walker2d, more training steps need to be shown.
- Robustness of the algorithm: are the results in Figure 1 derived from different random seeds? If so, how many of them?
- The experiment is only conducted in 4 control domains. The authors may consider to include more task domains, e.g., swimmer, and humanoid as well.
- Are the steps in Figure 1 correspond to only real simulator steps or total steps including imagined ones?
- The curves in Figure 2 are not shown with an uncertainty plot.

**Experience Assessment:**

I have read many papers in this area.

**Review Assessment: Checking Correctness Of Derivations And Theory:**

I carefully checked the derivations and theory.

**Review Assessment: Checking Correctness Of Experiments:**

I carefully checked the experiments.

**Review Assessment: Thoroughness In Paper Reading:**

I read the paper thoroughly.

---

> ### Author Response · Authors · 2019-11-15
> **Response to the comments**
>
> Thanks for the comments.
>
> Comments on the experiment.
> Response:We add results on Swimmer and Reacher in our paper.
>
> Comments on  motivation and novelty.
> It seems that the reviewer thinks S2VG is just “SVG+ entropy term”. However, it is not the case. Let me highlight our motivation [Section 1, paragraph 3 ] and main difference between S2VG with SVG again which has been discussed in [Section 1, paragraph 4].
> 1.	In fact, we propose our algorithm from the perspective of model-based RL. See [Section 1 , paragraph 3]. We embed the model in the policy update because recent state-of-the-art model-based RL algorithm first generates the imaginary data (from the model) and then trains the agent with model-free RL. That is not efficient since they know the model but do not use the model directly.
>
> 2.	The way to apply the model is totally different from SVG. SVG just utilizes the real data to update the policy, while S2VG uses the data generated from model (see details on our derivation and equation 11). Such difference is quite important. Using data from the model would reduce the number of interactions with the environment. One evidence is that most recent state-of-the-art model-based RL algorithm (e.g. SLBO) used imaginary data to train agent with model-free RL(e.g. TRPO) .
>
> 3.	In S2VG, we avoid the importance sampling in the policy update by sampling action from the policy and getting the transition according to the learned model .  In SVG, it has to introduce the importance sampling when it applies off-policy learning, since it just uses real data. Such way would introduce large variance due to the importance sampling.
>
> Another merit of our S2VG is that it is more computationally efficient than other state-of- the-art model-based RL. We can finish one trail of most of experiments in our paper around one hour on a laptop. We encourage reviewer to try our code.
>
>
> Comments: Robustness of the algorithm: are the results in Figure 1 derived from different random seeds? If so, how many of them?
> Response: The results shown in Figure 1 are tested based on 5 trails using 5 different random seeds and initialized parameters. We clarify that in the new version of the paper.
>
> Comments: Are the steps in Figure 1 correspond to only real simulator steps or total steps including imagined ones?
> Response:The steps in Figure 1 correspond to real simulator steps.
>
> Comments on the value expansion.
> Response:H here is the time step of value expansion using real data. \tau is the training tuple and \tau_0 is the initial training tuple. We agree that the definition should be clearly indicated. We clarify this in the new version of this paper.
>
> Comments on the “initial tuple is from real data”.
> Response: We clearly described the usage of real transition, imaginary transition, and its relationship with value expansion in Section 3.2, Paragraph 1 of the original version of the paper.  "Note that when H = 1, it reduces to the case where just real data is used."
>
> Comments : Up to how many roll-out step k could the model reliably work upon should be clearly stated.
> Response: Based on the definition in Equation 7, the roll-out of step k is from t to H-1.

---

### Decision · Program_Chairs · 2019-12-19

**Decision:**

Reject

**Comment:**

The authors consider improvements to model-based reinforcement learning to improve sample efficiency and computational speed. They propose a method which they claim is simple and elegant and embeds the model in the policy learning step, this allows them to compute analytic gradients through the model which can have lower variance than likelihood ratio gradients. They evaluate their method on Mujoco with limited data.

All of the reviewers found the presentation confusing and below the bar for an acceptable submission. Although the authors tried to explain the algorithm better to the reviewers, they did not find the presentation sufficiently improved. I agree that the paper has substantial room for improvement around clarity. Reviewers also asked that experiments be run for more time steps. I agree that this would be an important addition as many model-based reinforcement learning approaches perform worse asymptotically model free approaches and it would be interesting to see how the proposed approach does. A reviewer pointed out that equation 2 is missing a term, and indeed I believe that is true. The authors response is not correct, they likely refer to an equation in SVG where the state is integrated out. Finally, the method does not compare to state-of-the-art model-based approaches, claiming that they use ensembles or uncertainty to improve performance. The authors would need to show that adding either of these to their approach attains similar performance to state-of-the-art approaches.

At this time, this paper is below the bar for acceptance.